



# A new smog chamber system for atmospheric multiphase chemistry study: design and characterization

Taomou Zong[1], Zhijun Wu[1,2,*], Junrui Wang[1,3], Kai Bi[4], Wenxu Fang[1], Yanrong Yang[1], Xuena Yu[1], Zhier Bao[5], Xiangxinyue Meng[1], Yuheng Zhang[1], Song Guo[1,2], Yang Chen[5], Chunshan Liu[6], Yue Zhang[7], Shao-Meng Li[1], Min Hu[1,2]

[1]State Key Joint Laboratory of Environmental Simulation and Pollution Control, College of Environmental Sciences and Engineering, Peking University, Beijing 100871, China

[2]Collaborative Innovation Center of Atmospheric Environment and Equipment Technology, Nanjing University of Information Science and Technology, Nanjing 210044, China

[3]Laboratory of Atmospheric Observation Supersite, School of Environment and Energy, Peking University Shenzhen Graduate School, Shenzhen 518055, China

[4]Beijing Key Laboratory of Cloud, Precipitation and Atmospheric Water Resources, Beijing, 100089, China

[5]Research Center for Atmospheric Environment, Chongqing Institute of Green and Intelligent Technology, Chinese Academy of Sciences, Chongqing, 400714, China

[6]Beijing Convenient Environmental Tech Co. Ltd., Beijing 101115, China

[7]Department of Atmospheric Sciences, Texas A&M University, College Station, TX 77843, United States

*Correspondence to*: Zhijun Wu (zhijunwu@pku.edu.cn)

**E-mail lists:**
Taomou Zong (zongtaomou@pku.edu.cn)
Zhijun Wu (zhijunwu@pku.edu.cn)
Junrui Wang (18845725921@163.com)
Kai Bi (bikai_picard@vip.sina.com)
Wenxu Fang (343299989@qq.com)
Yanrong Yang (yyr2020@stu.pku.edu.cn)
Xuena Yu (99784925@qq.com)
Zhier Bao (baozhier@cigit.ac.cn)
Xiangxinyue Meng (mxxy96@126.com)
Yuheng Zhang (zhangyh@stu.pku.edu.cn)
Song Guo (guosong@pku.edu.cn)
Yang Chen (chenyang@cigit.ac.cn)
Chunshan Liu (bjkwnt@163.com)
Yue Zhang (Yuezhang@tamu.edu)
Shao-Meng Li (shaomeng.li@pku.edu.cn)
Min Hu (minhu@pku.edu.cn)



**Abstract.** Multiphase chemistry is an important pathway for the formation of secondary organic aerosols
in the atmosphere. In this study, an indoor 2 m³ Teflon chamber system (Aerosol multIphase chemistry
Research chamber, AIR) was developed and characterized to specifically simulate atmospheric
multiphase chemistry processes. The temperature and humidity controls, diurnal variation simulation,
and seed particle generation unit in this chamber system were designed to meet the needs of simulating
multiphase atmospheric chemical reactions. The AIR chamber is able to accurately control temperature
(2.5 ~ 31 ±0.15 °C) and relative humidity (RH < 2 % ~ > 95% ±0.75%) over a relatively broad range.
In addition, an RH regulation module inside the chamber was designed to simulate the diurnal variation
of ambient atmospheric RH. The aerosol generation unit is able to generate pre-deliquescent seed
particles with an organic coating across a wide range of phase states or morphologies. The organic
coating thickness of the aerosols within the chamber can be precisely controlled through adjusting the
condensation temperature, further helping to elucidate the roles of seed particles in multiphase chemical
reactions. The inner walls of the AIR chamber are passivated to reduce the wall loss rates of reactive
gases. Yield experiments of α-pinene ozonolysis with and without seed particles combined with a box
model simulation demonstrate the high-quality performance of secondary aerosol formation simulation
using the AIR chamber.

**1 Introduction**

Smog chamber is a mainstream tool in chemical laboratory studies to simulate the formation and
evolution of air pollutants (Batchvarova et al., 2006; Chen and Lelevkin, 2006; Kolev and Grigorieva,
2006; Mocanu et al., 2006; Tolkacheva, 2006) and reveal the parameterization or mechanisms of
atmospheric processes (Wenger, 2006; Olariu et al., 2006; Bejan et al., 2006; Mellouki, 2006; Barnes,
2006; Albu et al., 2006; Carter, 2006; Rudzinski, 2006; Zielinska et al., 2006). Chamber simulations have
irreplaceable advantages over other laboratory methods such as oxidation flow reactors (Kang et al.,
2007; Lambe et al., 2015; Corral Arroyo et al., 2018; Cosman and Bertram, 2008) and bulk solution
experiments (Brunamonti et al., 2015; Turšič et al., 2003; Pratap et al., 2021; Fleming et al., 2020; Mekic
et al., 2019) in tracking atmospheric transformation processes and understanding kinetic processes.
The development of chambers is closely related to advances in atmospheric chemistry research. Starting
with studies of photochemical smog in Los Angeles in the 1940s (Haagensmit, 1952) and continuing to
the 1970s, chambers were designed primarily to study the formation of ozone (Akimoto et al., 1979;
Carter et al., 1982) as well as the chemistry of volatile organic compounds (VOCs) and $NO_x$ (Morriss et
al., 1957) in the atmospheric boundary layer. With the development of submicron particle measurement
techniques, chambers were further used in secondary organic aerosol (SOA) formation studies from the
1980s leading to numerous important scientific discoveries (Hidy, 2019; Odum et al., 1996; Odum et al.,
1997; Griffin et al., 1999; Paulsen et al., 2005; Rollins et al., 2009; Hu et al., 2014; Wang et al., 2014).



Since the beginning of the 21st century, many chambers have been built or upgraded to address integrated
atmospheric scientific questions, including $PM_{2.5}$ pollution (Johnson et al., 2004; Hallquist et al., 2009;
Hurley et al., 2001), reaction kinetic parameters, mechanisms of VOC oxidation intermediates (Brauers
et al., 2003; Bohn et al., 2004; Ren et al., 2017), as well as multiphase processes (Warneke and C., 2004;
Pöschl and Shiraiwa, 2015; Liu and Abbatt, 2021; Franco et al., 2021).
In recent years, multiphase chemistries have been invoked to explain the bursting growth of particles (Su
et al., 2016; Wang et al., 2016; Su et al., 2020) and physicochemical processes of SOA formation under
high ion strength conditions in the atmosphere (Cheng et al., 2015; Su et al., 2020; Liu et al., 2021).
Atmospheric multiphase processes can undergo different reaction pathways that are influenced by
different environmental conditions (e.g., light, temperature, and relative humidity (RH)) and aerosol
physicochemical properties including aerosol liquid water content (ALWC), aerosol phase state, and
morphology (George and Abbatt, 2010; Davidovits et al., 2011; Abbatt et al., 2012; Ziemann and
Atkinson, 2012; Herrmann et al., 2015; Ravishankara, 97; George et al., 2015; Su et al., 2020). Thus, a
precise control of such parameters in a chamber system is vital for simulating atmospheric multiphase
chemistry. Different from outdoor chambers (Leone et al., 2010; Stern et al., 1987; Pandis et al., 1991;
Johnson et al., 2004; Martin-Reviejo and Wirtz, 2005; Rollins et al., 2009; Cocker et al., 2001; Peng et
al., 2017), indoor chambers are usually equipped with artificial light sources (Takekawa et al., 2003;
Carter et al., 2005; Paulsen et al., 2005), that can provide controllable irradiation for the simulation of
multiphase processes. Compared to large chambers (Brauers et al., 2003; Leone et al., 1985; Pandis et
al., 1991), temperature and RH inside small chambers can achieve faster equilibria and provide a more
precise simulation of parameters such as diurnal RH change and ALWC (Takekawa et al., 2003; Carter
et al., 2005; Paulsen et al., 2005; Wang et al., 2014; Bin Babar et al., 2016), thus improving
reproducibility and efficiency when conducting experiments. Adversely, the wall loss effects are more
significant for small chambers (Carter et al., 1982; Carter and Lurmann, 1991; Dodge, 2000). As studies
showed evidence that the morphology and phase state of aerosol particles play important roles in the
atmospheric multiphase chemistry processes (Virtanen et al., 2010; Berkemeier et al., 2016; Wang et al.,
2015a; Reid et al., 2018), focused chamber studies on multiphase chemistry require additional steps to
control the morphology and phase state of seed particles in chamber design(Faust et al., 2017; Zhou et
al., 2019; Zhang et al., 2018; Zhang et al., 2019).
In this study, we designed and built a new indoor 2 m³ Teflon chamber system (Aerosol multIphase



process Research chamber, AIR) with a focus on accurately simulating atmospheric multiphase processes.
The temperature and RH inside the AIR chamber were precisely controlled to within ± 0.15 ℃ and ±
0.75 %, respectively. A quantitative manipulation of the RH cycle was designed to simulate the diurnal
variations in ambient RH. The seed generation subsystem, including an inorganic particle pre-
deliquescence unit and an organic-coating unit, was designed to manipulate the aerosol phase state and
organic-coated morphology. A series of experiments were conducted to characterize the spectral
distribution and photolysis parameters of light sources, temperature, RH, wall loss behaviors of gas and
particles, and particle morphology. Additionally, a series of experiments involving the oxidation of α-
pinene with seed particles were conducted in the AIR chamber to demonstrate the effectiveness of the
chamber in simulating atmospheric multiphase chemistry.
**2 Facility**
Figure 1 displays the schematic design of AIR chamber system, and the real picture of the reactor bag
and enclosure system are shown in Fig. S1. The chamber system includes the 2 m$^3$ fluorinated ethylene
propylene (FEP) Teflon film (75 μm, Du Pont, USA, light transmission ≥ 93%) reactor and the associated
temperature and RH control, artificial light sources, zero air injection and humidification, gaseous/liquid
precursor injection, seed aerosol generation, and the instrument-optional detection components. To
achieve a precise control of thermodynamic parameters and aerosol morphology when simulating
atmospheric multiphase chemistry processes, the temperature inside the reactor is precisely controlled to
within ± 0.15 ℃. An RH regulation module is designed and built to simulate the ambient RH diurnal
variation, which is capable of changing the RH in the reactor at a time scale of half an hour. In addition,
a pre-deliquescing device and a coating device are custom-built to couple to the seed aerosol generation
component, for manipulating the phase state (metastable aqueous or solid) and core-shell morphology
(1 % ~ 12 % shell thickness) of seed aerosols. The detailed description of each system is shown in Section

126    2.1-2.4.



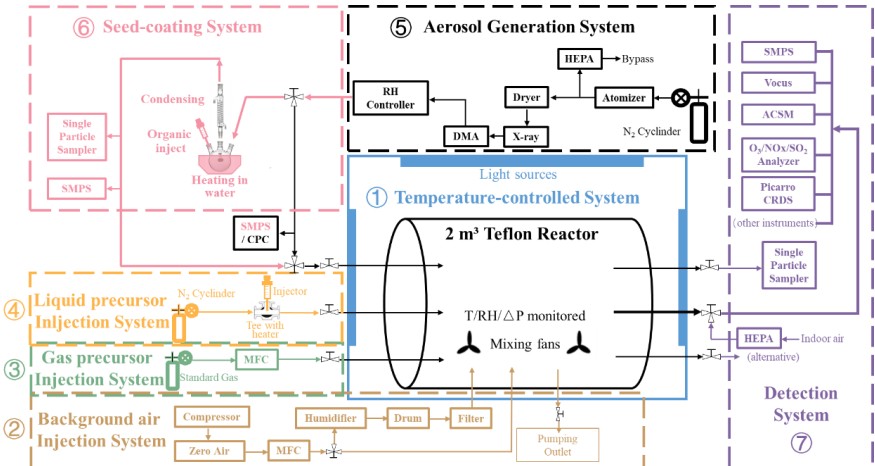

**Figure 1. Schematic diagram of AIR chamber system.**

## 2.1 The reactor and enclosure

The Teflon reactor is a 2 m³ horizontal cylinder (1.2 m in diameter, 1.8 m in length). It is fixed on a stainless-steel frame with four ridges firmly adhered on the Teflon air bag (Fig. S1), so that the variable volume of the reactor during sampling is adequate (this chamber system is designed to operate in Batch Mode). As to each circle side of the cylinder, three stainless steel tubes are threaded through the Teflon film to act as the inlets (for injecting seeds and liquid phase precursors) or sampling outlets for the detection system, respectively. The interface between each tube and the film is sealed by a Teflon flange and a perfluorinated O-ring. At the bottom inside the reactor, two magnetic-levitation fans (patent number: 2019213329392, Beijing Convenient Environmental Tech Co. Ltd.) are equipped, with four speed levels (1000, 1350, 1700, 2000 rpm). A temperature and RH sensor (HMP110, Vaisala, Finland) and a differential pressure sensor (MSX-W10-PA-LCD, Dwyer, America) are also equipped at the bottom inside the reactor.

The rectangular enclosure (2.4 × 1.6 × 2.3 m, L, W, H) of the reactor is temperature-controlled by a circulation system. The indoor air is introduced from the top of the enclosure and exhausts through the bottom. The chiller power is constant, while the heating power is controlled through a proportional-integral-derivative (PID) feedback. Forty black lights (1.2 m, 40 W, Bulb-T12, GE, USA) are fixed on the inner wall of the enclosure as light sources for atmospheric process simulation. The number and position of these lights in work can be controlled by the system computer, so that the light intensity can be variable in experiments. Specular insulated material (SUS304, stainless steel, 8K, mirror plate) is used





as the enclosure inner wall so that the irradiation inside the reactor can be homogeneous. One side of the
enclosure is a double door for entering and reactor maintenance.

**2.2 Cleaning and humidifying system**

The background gas in the reactor is from the indoor air. An air compressor (FOHUR, FH-50L)
compresses the indoor air into a zero-air generator (Aadco, 737-14-A-CH4-240) for purification,
removing airborne contaminants such as particulate matters, hydrocarbons, water vapor, NOx, $O_3$ and
$SO_2$ to produce zero air (RH can be dried to < 2%, and the background concentrations of other
contaminants are displayed in Table S2). Then, with the control of a mass flow controller (MFC,
HORIBAMETRON, S4832/HMT), zero air is fed into the reactor through a 1/2" stainless steel tube
(sealed at the bottom interface by a 304 stainless steel flange) at a flow rate of ≤ 50 L/min (to ensure the
cleaning efficiency of the zero-air generator is sufficient), acting as the background gas and cleaning gas
for the reactor. At the same time of feeding into the cleaning zero air, a pump beside the chamber system
will exhaust the air from the reactor with a flow rate of 20 L/min to accelerate the gas exchange. The
positive differential pressure inside the reactor is monitored. When the differential pressure reaches 30
Pa, the MFC will stop the zero-air feed, and when the value falls below 20 Pa, zero air feed will restart.
This is designed to avoid damaging the Teflon film of the reactor during cleaning.
The zero air is also used as humidifying gas. When switching to the humidification mode, the zero air
will go into a humidification tank filled with deionized water (Milli-Q, 18MΩ) switched by a three-way
valve, generating humidified zero air. Then, the humidified air flows through a filter (Waterman, HEPA)
to remove the water droplet, and injects into the reactor to humidify. During the humidifying, the exhaust
pump mentioned above keeps working. The flow rate of the humidified zero air (20 ~ 25 L/min) is set to
be slightly higher than the exhausting rate for fast reaching the target RH inside the reactor.

**2.3 Precursor injection system**

According to the phase state of precursor reagents, the precursor injection system of this chamber system
contains two types. One is used for the injection of gaseous precursors. Standard gas cylinders containing
reactive gas (such as $SO_2$, $NO_2$, $NH_3$, HCHO, etc.), inject relevant gaseous precursors into the reactor at
a set flow rate and injecting duration under the control of a computer-connected MFC. The oxidant $O_3$ is
produced through the decomposition of $O_2$ (from a standard $O_2$ cylinder) exposed to the 185 nm UV light.


After flowing through the MFC, the gaseous species enter the reactor via a stainless-steel tube at the
bottom of the chamber.
The other type is used for the injection of liquid precursors. Note that, the liquid precursors here mean
the species is in liquid phase before injected into the reactor, but should be gaseous after injecting into
the chamber, such as α-pinene standard solvent. A tee (the inlet on the left side of the chamber, as shown
in the 'Liquid precursor Injection System' in Fig. 1) is fitted in the pipeline before the liquid precursors
entering the reactor, with a 1 mm thick silicone membrane clamped to the right-angled end. The specific
amount of the liquid precursors is taken with a microsyringe, penetrating the silicone membrane and
slowly injected into the tee. At the same time, pure $N_2$ is used as the carrier gas to vaporize the liquid
precursor and carry it into the reactor under a specific gas cylinder pressure (0.25 MPa). After injection,
$N_2$ is continuously purged for 60 seconds to ensure that no liquid precursors remain in the pipeline.
**2.4 Seed generation system**
The seed aerosol generating system is a complex subsystem of AIR chamber system designed in this
study. In addition to the common aerosol generation device, this study couples an RH-controlling device
and a coating device to control the phase state and morphology of the seeds for supporting the simulation
of atmospheric multiphase processes.
Commonly, the species used to generate the seed particles (typically dissolved inorganic salts such as
ammonium sulfate and sodium chloride) are first dissolved in deionized water (Milli-Q, 18 MΩ) and
then generate a solution. Then, it is atomized as humid aerosol flow by an atomizer (TSI 3076) with $N_2$
blowing. Passing through a Nafion tube (PERMA PURE, MD-700-24F-3), the humid flow is dried and
forms dry polydisperse seed aerosols. The drying is realized by pumping the air at the outer layer of the
Nafion tube to a negative pressure (~20 kPa). It is tested that, within the range of the aerosol generation
flow rate ($\leq$ 3 L/min), the RH of the aerosol flow can be dried to below 30 %. An X-ray neutralizer and
DMA (DMA, Model 3082, TSI, Inc., USA) are optional, for selecting monodisperse aerosols from the
polydisperse aerosol flow (flow rate ratio of sheath flow to aerosol flow is controlled between 5:1 and
10:1), to support monodisperse experiments.
Besides, an RH controlling device is designed in this study to pre-deliquesce the generated dry seeds that
forming metastable seed aerosols. As shown in Fig. S9, $N_2$ is used as the initial gas, which is then divided
into two paths, one is the dry $N_2$, and the other goes through the deionized water (Milli-Q, 18MΩ, heated



to 45 °C) to act as the wet gas. The flow rate of each path is controlled by an MFC (GAS TOOL
INSTRUMENT, GT 130MAX). Then the two flows mix into one as the humidifying gas and enter the
outer layer of a Nafion semi-permeable tube (PERMA PURE, MD-700-24F-3). The flow with seed
aerosols goes through the inner layer of the Nafion tube and then is humidified. The RH of the humidified
flow is detected by an RH sensor (HYGROCLIP2, HC2A-S). The two MFCs of each flow path and the
RH sensor are connected to a computer and controlled by a Labview program with PID feedback.
Through the two MFCs adjusting the ratio of the flow rates of the dry and wet flow path, the RH of seed
aerosol flow is controlled. This device has been tested to enable rapid changes in RH between 5 % and
90% within 5 mins, and the RH variability can be within ± 0.2 %.
In order to investigate the effect of aerosol coating on atmospheric multiphase process, a device is
designed in this study to generate a thickness-controlled and species-known coating on the generated dry
monodisperse seed aerosols. The constitution of the coating device is shown in Fig. S10. This device
consists of a water bath (Changfeng, HW.SY11-KP1), a three-necked flask (250 mL, 19#-24#-19#), a
condensing glass tube (30 cm, 24#), and a thermostatic bath (BiLon, SC-05B). The organic species (~
400 μL) with low volatility (saturated vapor pressure in the order of $10^{-4}$ ~ $10^{-5}$ mmHg at room
temperature) used to form coating is set at the bottom of the three-necked flask, which is heated in the
water bath to evaporate the organic vapor. The dried seed aerosol flow enters through the side port of the
three-necked flask, and then carries the hot organic vapor into the condensing tube (condensing
temperature is controlled at 20 °C by the thermostatic bath in this study). Due to the reduced temperature,
the saturated vapor pressure of the organic drops, and the organic vapor will preferentially condense on
the surface of seed aerosols that forming a coating.
**2.5 Detection system**
As shown in Figure 1, three stainless steel tubes are fixed on the right side of the reactor to act as sampling
outlets. The middle steel tube of them is 3/8 " in size and acts as the main sampling tube, connected to a
3/8 " stainless steel three-way plug valve. One outlet of the plug is attached to a HEPA filter, and the
other outlet is attached to the line to sampling instruments. This design allows a quick sampling switch
between indoor air and the reactor. The other two stainless steel tubes are both 1/4 " and are used as
auxiliary sampling outlets (e.g. temporarily collect single particle samples for a few minutes).
An SMPS system (a DMA, Model 3082, and a CPC, Model 3772, TSI, Inc., USA), and a CPC (Model





3750, TSI, Inc., USA) downstream of the seed generation system, are the standing instruments for the
chamber system, used to measure the particle number size spectrum distribution and particle total number
concentration in the reactor, respectively. Other instruments are optional according to the specific
research aim, and typically the total sampling flow rate should be lower than 6 L/min.
The other detection instruments involved in this study, include the instruments for gaseous species
detection (Thermo Scientific gas analyzer (Model 43i-TLE for SO2, Model 42i-TL for NOx, Model 49i
for O3, Model 48i-TLE for CO), Picarro cavity ring-down spectroscopy (Picarro CRDS, G2401) for $CO_2$
and $CH_4$, Summa Canister (SILONITE, 1869) and GC-MS (Agilent, 7890A/5975C) for non-methane
hydrocarbon (NMHC)), instruments for particulate species detection (Time-of-Flight Aerosol Chemical
Speciation Monitor (ToF-ACSM, Aerodyne)), and instruments for volatile organic compounds (Vocus
Proton-Transfer Reaction Time-Of-Flight Mass Spectrometry (Vocus-PTR-TOF-MS, Vocus S, Tofwerk),
shorted as Vocus).
The sampling flow rate of each instrument is calibrated before each experiment. For Thermo Scientific
instruments and Vocus, a single standard concentration is tested at each experiment, to act as a basis for
instruments status verification and data quantification. For the data collected by ACSM, the calibration
is performed based on the mass concentration calculated from SMPS data.
**3 Characterization of the AIR chamber**
A series of experiments were carried out to evaluate the performance of this chamber system, including
leakproofness, sample-volume support, background concentrations, mixing performance, light
characteristics, temperature and RH control, gas and particle wall loss, as well as characterizations of
aerosol particles with the core-shell morphology. All the instruments for measurement are included in
Section 2.5.
**3.1 Fundamental parameters**
Leakproofness of the reactor was characterized by the positive pressure difference between the air inside
the reactor and the ambient air and the change in the total number concentration of background particles
inside the reactor. When the reactor was filled with zero air, the positive pressure difference inside the
reactor was maintained at > 3 Pa for more than 25 hours (Fig. S2a), then slowly decreased to ~ 0.5 Pa
after several days. When the air inside the reactor was sampled at a flow rate of 5 L/min, the positive



pressure difference decreased to zero after 2 hours, and then total particle number concentration slowly
increased from ~ 0 cm$^{-3}$ to a final < 10 cm$^{-3}$ in ~3.5 hours (Fig. S2b). This concentration is negligible for
a particle number concentration of $10^3 ~ 10^4$ cm$^{-3}$ that are usually used in experiments. Moreover, this
chamber system is designed to operate in batch mode, and the reactor can provide a sampling volume of
1000 ~ 1200 L (Fig. S3) and a sampling time of more than 3 hours at a total sampling flow rate of 5~6
L/min. The results above indicate that the system leakproofness is reliable for further experiments.
The reactor background was also characterized after repeated cleaning with zero air. As shown in Fig.
S4, the background particle total number concentration was < 1 cm$^{-3}$, and increased only to 4 cm$^{-3}$ with
the mixing fans turned on. Irradiation slightly increased the background particle concentration but still
only to < 10 cm$^{-3}$, which is negligible when compared with normal reaction conditions. Table S1 shows
the background concentrations of chemical species in AIR chamber reactor under dry and high RH
conditions. Compared with data reported for other chambers (White et al., 2018; Bin Babar et al., 2016;
Wang et al., 2014; Platt et al., 2013; Carter et al., 2005; Chen et al., 2019b), the background
concentrations of gaseous pollutants including $SO_2$, NOx, $O_3$ and CO in the reactor were comparable or
lower for the AIR chamber. The background concentration of total non-methane hydrocarbon (NMHC)
was higher than literature values due to the presence of chemically inert $CHClF_2$ (half of the total NMHC
concentration), which originates from the indoor refrigeration system and is hard to eliminate within the
zero-air generation system. Nevertheless, this species does not interfere with the reactions under most
experimental conditions. The reactor can be cleaned to background levels with a volume of zero air >5
times that of the reactor (Table S2) after each experiment. The cleaning process can be completed in less
than 9 hours, as shown in Section 2.2.
The mixing performance of the injection into the reactor was examined using $NO_2$ concentration and
total particle number concentration as tracers (Fig. S5). The mixing time to uniformity was 5 minutes
without running fans and less than 1 minute with the fans on. Furthermore, the mixing time was
independent of the fan speed.
**3.2 Light source characterization**
The reflective inner wall (SUS304, stainless steel, 8K, mirror plate) of the AIR chamber is equipped with
40 UV lamps (1.2 m, 40 W, Bulb-T12, GE, USA) to provide irradiation during the experiments. There
are 10 lamps on the left, right, back, and top of the wall, respectively, and each lamp can be turned on or



off separately by the control system, so that the light intensity in experiments varies from 2.5% to 100%
intensity. These light sources can also be replaced by lamps with different emission spectra to provide a
variety of irradiation conditions.
For current light sources, a portable UV spectrometer (StellarNet Inc., Tampa FL, USA) was used to
characterize the irradiance spectrum in the reactor (Fig. S6). The irradiance is mainly distributed in the
range of $360 \sim 390$ nm, peaking at 370 nm, which is within the range of peak irradiance of UV lights
used in other indoor chambers ($340 \sim 371$ nm) (Wang et al., 2014; Ma et al., 2022; Bin Babar et al., 2016;
Chen et al., 2019b; Lane and Tang, 1994; Thuner et al., 2004). Another small peak appears at 405 nm,
which is convenient for directly checking the status of the lamps.
The photolytic rate constant for $NO_2$ can be used to characterize the irradiation intensity. Previous
literature (Wang et al., 2014; Bin Babar et al., 2016; Ma et al., 2022) often characterize irradiation
intensity through the photolytic rate constant of $NO_2$ (J_$NO_2$), calculated through the steady-state
concentrations of NOx and $O_3$ (Atkinson et al., 2004). This study mainly used a spectrometer, namely
the Jvalue instrument (AVANTES, AvaSpec-ULS-TEC-EVO), to measure the irradiance and directly
calculate the photolytic rate constants of a few important species in atmospheric photochemistry. Notably,
the Jvalue instrument was also calibrated using the J_$NO_2$ values derived from the steady NOx-$O_3$
concentration under several light schemes to correct for the geometry defect of the Jvalue instrument
when placed inside the AIR chamber. The calibration factor of the traditional J_$NO_2$ method is 1.49 ±
0.06. As shown in Table S3, the current light source is more suitable for the photolysis of HONO and
$NO_2$ (photolytic rate constants on the order of $10^{-4} \sim 10^{-3}$ s$^{-1}$). However, the photolysis of HCHO, $H_2O_2$,
and $O_3$ is slow (photolytic rate constants on the order of $10^{-8} \sim 10^{-7}$ s$^{-1}$). The J_$NO_2$ maxima of other
chambers are usually in the range of $2 \sim 9 \times 10^{-3}$ s$^{-1}$ (Chen et al., 2019a; Li et al., 2017; Wang et al., 2014;
Bin Babar et al., 2016; Ma et al., 2022). In comparison, J_$NO_2$ due to the light source in the AIR chamber
is $4.10 \times 10^{-3}$ s$^{-1}$, close to the median value of the other chambers. Moreover, the photolytic rate constant
of HONO due to the light source in this chamber (J_HONO at the level of $10^{-4}$ s$^{-1}$) is comparable to or
slightly higher than the value of HONO photolysis in the ambient atmosphere in China (J_HONO at the
level of $10^{-5} \sim 10^{-4}$ s$^{-1}$) (Zheng et al., 2020).
When only lamps on two sides of the AIR chamber were turned on (four schemes with 20 lights on, noted
as 'only back/top', 'left and right', 'odd' and 'even' in Table S3), the photolytic rate constants in the
reactor under different configurations were almost the same (J_HONO = $5.10 \pm 0.12 \times 10^{-4}$ s$^{-1}$, J_$NO_2$ =



$2.16 \pm 0.05 \times 10^{-3}$ s$^{-1}$), and nearly equal to half of that with all 40 lights on. In addition, the photolytic
rate constant of the scheme 'left and right' (40 lights) was the sum of that of 'only left' (20 lights) and
'only right' (20 lights). These results indicate that the irradiation in the reactor is uniformly distributed.
Notably, because the measurement interface of Jvalue was a little biased to the left during detection, the
value for 'only left' was higher than that for 'only right'.
**3.3 Performance of temperature and RH control**
The temperature and RH in the reactor are measured by a high-accuracy sensor (HMP110, Vaisala,
Finland). Detailed descriptions of temperature and RH control are given in Section 2.1 and 2.2. The
accuracy for RH of this sensor is shown by its measurement error of < 1% from that measured by a
hygrometer (chilled mirror hygrometer, Edgetech Instrument, USA), with an $R^2 > 0.99$. The temperature
in the reactor can be stably controlled in the range of 2.5 °C ~ 31 °C, and the control range of RH is < 2%
~ > 95%. The fluctuations in the temperature inside the reactor are within $\pm 0.15$ °C of any set temperature,
and the corresponding RH fluctuations for RH > 80 % are within $\pm 0.75$ %. The stability achieved with
the temperature and RH controls across a wide range of temperatures is shown in Table S4. The
illumination of lamps raises the lowest achievable temperature by 3 °C for every 10 lights on. However,
the illumination of the reactor does not affect the stability of temperature and RH inside the reactor. When
the set temperature is close to room temperature (20 °C in Table S4), the fluctuation is < 0.1 °C,
demonstrating a more accurate temperature and RH control performance compared with other chambers
(Table S5) (Wang et al., 2014; Wu et al., 2007; Bin Babar et al., 2016; Ma et al., 2022; Wang et al.,
2015b). Sampling operation (lasting more than 3 hours with flow rate at 5 L/min, Fig. 2) does not
significantly affect the stability of temperature and RH control either, which also indicates the permeation
and wall loss of water molecules do not affect a lot.
In order to simulate the diurnal variations in ambient air temperature and RH, a proportional-integral-
derivative (PID) feedback controlling function was designed. The RH in the reactor can reach the target
RH by controlling the temperature. After receiving the target RH input, the control program calculates
the stepwise theoretical RH value at each time increment and the corresponding temperature control steps
based on current temperature and RH in the reactor. This calculation is also adjusted in real-time to
optimize the gradual change of RH. Figure S7 demonstrates two examples to show alternate linear change
and constant control of RH. The RH can reach the set value within a few hours with fluctuations < 0.75%.





This function performs even better at low temperatures, suggesting the potential of using this chamber
system to simulate diurnal variations of RH in the ambient atmosphere in wintertime.

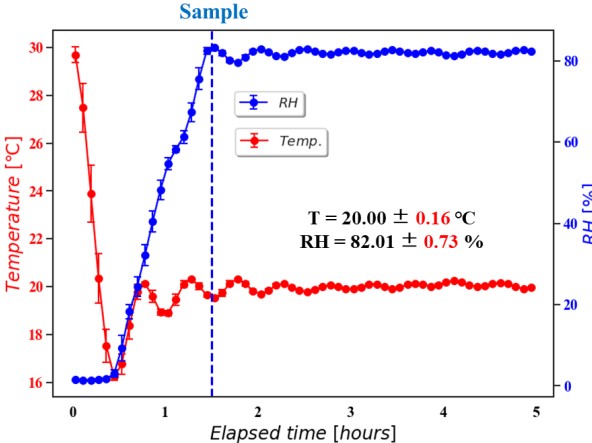


**Figure 2. Stability of temperature and RH control in the reactor during sampling.**
**3.4 Wall loss of gas and particles**
The wall loss process is considered as a first-order kinetic process, in that the decay rate of a
concentration is proportional to the concentration:
$$\frac{dC(t)}{dt} = -k * C(t)$$    (1)
where $C(t)$ is the species concentration at time t, and k is the wall loss rate constant (in units: $s^{-1}$ or min$^{-}$
$^{1}$). The wall loss rates of gaseous species such as $NO_x$ and $O_3$ in this study are shown in Table S6, the
values of which are lower than other small Teflon chambers (2 ~ 5 m$^3$) (Wu et al., 2007; Wang et al.,
2015b; Li et al., 2017; Bernard et al., 2016), as a result of passivation of the inner surface of the reactor
with 2 ppm $O_3$ for 3 days.
The wall loss rate constant k of particles is dependent on particle size (diameter, noted as $D_p$). Smaller or
larger particles often have higher k values (Crump and Seinfeld, 1981) due to higher diffusion or
sedimentation rates, respectively. The dependence of k values for particles with $D_p < 50$ nm is rarely
reported in previous chamber studies. This study demonstrates that the constant k decreases as a function
of decreasing $D_p$ when particles are smaller than 50 nm, which is also shown in Fig. S7 of Ma et al (Ma
et al., 2022). The $log_{10}(k)$ value for particles can be approximated with a segmented linear function of
$log_{10}(D_p)$ [93, 94]. In addition to the slopes to be determined, the inflection point $D_p$, where the loss trend



inverses, changes with different chambers. In this study, two inflection points are identified at 50 nm and
150 nm (Fig. S8). Furthermore, the k-Dp dependence has been reported to deviate in different
experiments even in the same reactor. This study found that such deviations can be corrected through an
up-and-down shift of the $\log_{10}(k)$-$\log_{10}(Dp)$ function curve. Even for deliquescent particles (RH = 90 %
in Fig. S8, the Dp of the x-axis represents the liquid particle diameters), this method still accurately
described the relationship between k and Dp ($R^2$~0.95) when considering the hygroscopic growth of the
particle size.
Another commonly used parameter to characterize the particle wall loss behavior in chambers is the total
volume wall loss rate constant ($k_v$). For small Teflon chambers of 2 ~ 3 $m^3$ in size (Takekawa et al., 2003;
Li et al., 2017; Liu et al., 2019), $k_v$ values typically range from 2.84 ~ 4.72 * $10^{-3}$ $min^{-1}$. The particle wall
loss is slightly higher in the chamber in this study, with the $k_v$ found to be 5 × $10^{-3}$ $min^{-1}$ (Table S7).
**3.5 Morphology of seed particle generation**
Seed particles are typically used to simulate aerosol formation by the multiphase chemistry pathway. The
AIR chamber is designed to couple to a subsystem for generating seed particles with different phase
states through pre-deliquescing, adopted from a previous study (Faust et al., 2017). A volatilizing-
condensing method is used to generate known-composition organic-coated inorganic particles in the AIR
chamber, with a detailed description in Section 2.4.
As shown in Figure 3, squalane is coated onto dry 200-nm monodisperse NaCl seed particles to produce
a core-shell morphology for the particles. The coating thickness is controlled by adjusting the water bath
heating temperature while maintaining a fixed condensation temperature of 20 ℃. Using the Clausius-
Clapeyron equation that describes the relationship between saturation vapor pressure and temperature,
as well as the Maxwell equation that describes the condensation growth rate of particle size under a
certain supersaturated vapor pressure, the coating thickness can be predicted in relation to the heating
temperature (Fig. 3a), to assess the feasibility of the selected coating species. The coating thickness is
calculated as half of the difference in peak Dp of the monodisperse particle size distribution before and
after the seeds are coated (Fig. 3b). For squalane, the device allows for a relatively accurate control of
coating thickness in the range of 5 to 25 nm (1 % ~ 12 % shell thickness). For organic species with similar
volatilities (saturated vapor pressure in the order of $10^{-4}$ ~ $10^{-5}$ mmHg at room temperature), the device
could provide similar control performance.

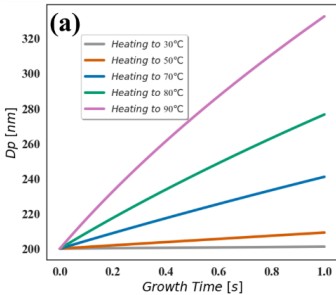
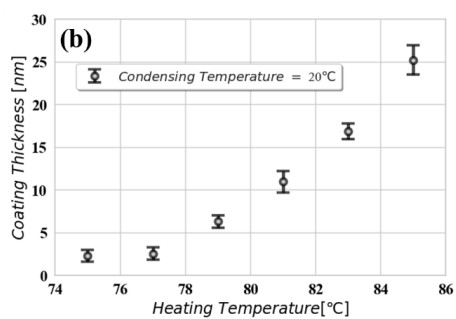


**Figure 3. Relationship between coating thickness on dry 200 nm NaCl seed and heating temperature in the coating device, with squalane as the coating species and 20 °C condensing temperature. (a) Theoretical estimation in different growth times. (b) Measured results by SMPS.**




**4 Applications in SOA generation——α-pinene ozonolysis researches**
**4.1 SOA yield of seed-absent experiments**
SOA are generated from α-pinene ozonolysis in the AIR chamber to evaluate its performance, with
experiment conditions given in Table S8 (NO.1 ~ 5). The key parameter Y, representing the yield of SOA,
is defined as:
$$Y = \frac{\Delta mo}{\Delta ROG} \tag{2}$$
where Δmo represents the total mass concentration of generated SOA, and ΔROG represents the total
mass concentration of reactive organic gas that was consumed in the reaction (specifically referring to α-
pinene in this study), with both units in μg/m³. SOA mass concentration was measured by a ToF-ACSM
(Section 2.5). The organic mass measurement was also corrected based on the particle size distribution
data from SMPS, where the α-pinene-derived SOA density was assumed as 1.3 g/cm³. This density value
is also used in many previous researches (Bahreini et al., 2005; Alfarra et al., 2006; Ma et al., 2022), but
higher than the unit density assumption used in some other chamber studies (Wang et al., 2011; Wang et
al., 2014; Bin Babar et al., 2016; Cocker Iii et al., 2001; Li et al., 2021; Zhang et al., 2015).
Odum et al (Odum et al., 1996) found that the two-product model reproduces well the non-linear
relationship between the SOA yield Y and the particulate organic mass concentration (mo):
$$Y = mo * \sum \frac{\alpha_i * K_{om,i}}{1 + mo * K_{om,i}} \tag{3}$$
where $\alpha_i$ and $K_{om,i}$ are the mass-based stoichiometric and partition coefficient for species i, respectively,
and mo is the total mass concentration of organic aerosol. Figure 4 shows the results of the two-product
model that fits the seed-absent SOA yield results in this study. The Odum model fits results from other
chamber studies are also shown in Figure 4 for comparison. Detailed model fitting parameters are shown
in Table S9. In contrast, Y in this study is a little higher than those in other small or medium-sized
chambers, which may be owing to the lower gas wall loss in our Teflon reactor (Section 3.4). The four
fitting parameters in this study, α1, α2, K1, K2, are 0.62479, 0.0326791, 0.0121589, 0.0121596,
respectively. K1 and K2 are close and are moderate values; however, α1 is significantly higher than those
in other chambers. Such higher value for α1 can be an indication of a lower volatilizing loss of the gas
phase intermediates within the AIR reactor compared with the other chambers. The good fitting from our
experiment indicates that the chamber system in this study is stable. These results imply a reliable
performance of our chamber system for experimental simulation studies of atmospheric secondary
transformation process.

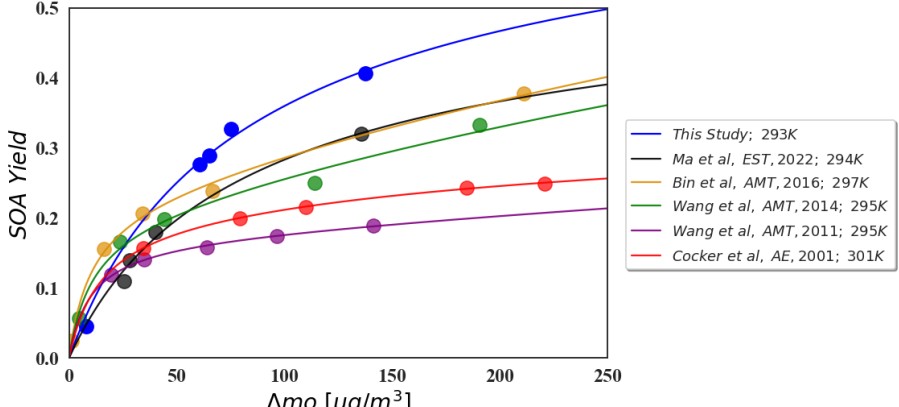


**Figure 4. Two-product model fitting curve of seed-absent α-pinene-derived SOA yield in this study and the**
**comparison with other literature results. The data of the blue line is from this study, and other data is obtained**
**from these references (Cocker Iii et al., 2001; Wang et al., 2011; Wang et al., 2014; Bin Babar et al., 2016; Ma**
**et al., 2022).**
**4.2 Effects of seed phase state on SOA yield**
The effects of different seed phase state on the yield of α-pinene-derived SOA were further investigated
using ammonium sulfate as the seed particles (Table S8, NO.6 ~ 8). Figure S11 shows the relevant
measured parameters during one reaction (e.g., experiment NO.8). The yields of all the experiments are
summarized in Fig. 5. In general, the yield in the presence of dry seeds is not significantly different from
that in the absence of seeds, consistent with the outcome of Odum et al (Odum et al., 1996). However,
in the presence of aerosol liquid water and ammonium sulfate seeds, the α-pinene-derived SOA yield is
reduced. This suppressing phenomenon is also reported by Cocker et al (Cocker Iii et al., 2001), which
may be related to the finding of Lutz et al (Lutz et al., 2019) that an inhibition of organic species
partitioning in the particulate phase exists at high sulfates level. However, to our knowledge, the
suppressing phenomenon above may not be common, that has only been reported in the α-pinene
ozonolysis system with ammonium sulfate seeds.
The subplot in Fig. 5 demonstrates the SOA yield at each elapsed time point in these experiments. Liquid
water can significantly promote the initial SOA yield and generation rate (Zhang et al., 2018), and our
results have reproduced this phenomenon (subplot in Fig. 5). However, the oxidation reaction proceeds,
it is observed that the SOA yield with liquid seeds decreases, and larger seed aerosol liquid water contents
produce greater decreases in the yield. These indicate the AIR chamber system facilitates the researches
of aerosol properties on atmospheric multiphase processes.

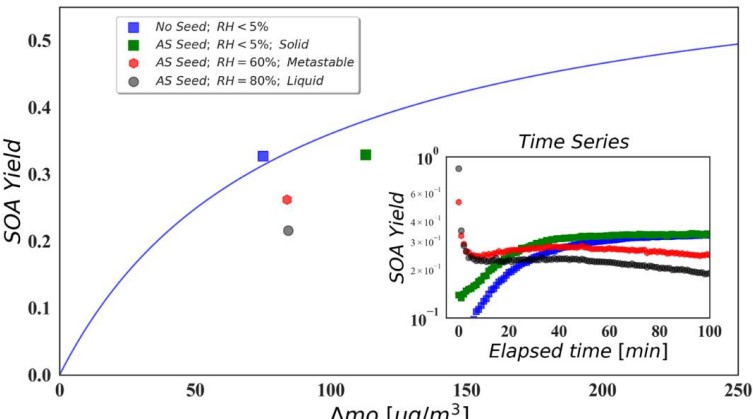


**Figure 5. Effects of phase state and liquid water content of ammonium sulfate seed particles on the SOA yield**
**of α-pinene ozonolysis (α-pinene = 60 ±13 ppb, O3 = 296 ±30 ppb). In the main plot, the blue line is the fitting**
**two-product curve from no-seed experiments data in this study, which is a replicate of the curve in Fig. 4. The**
**subplot shows the current yield since the initial time point of each experiment, where the blue points represent**
**the data of no-seed experiment, green points represent the data of solid seed experiment, red points represent**
**the data of metastable seed experiment, and grey points represent the data of liquid seed experiment.**
**5 Conclusions**
The reported special phenomena relying on specific particle properties are well reproduced in AIR



chamber benefitting from the seed phase state control, and the accurate temperature and RH control
facilitates the quantization of the effects of aerosol liquid water. Besides, compared to other chambers,
the manipulation of composition and thickness of organic coating could provide a more clarity surface
property. Broad temperature range, adjustable irradiation intensity, and the fast-responding RH cycle,
make this chamber system suitable for simulating diurnal ambient atmosphere in different seasons. These
performances of handling key parameters suggest the potential of this AIR chamber system for the
laboratory simulation of atmospheric multiphase processes.
**Data availability**
The data in this study are available from the authors upon request (zhijunwu@pku.edu.cn).
**Acknowledgements**
We thank the Beijing Convenient Environmental Tech Co. Ltd. for constructing the chamber.
**Financial support**
This research was financially supported by the National Natural Science Foundation of China

478    (41875149).

**Author contributions**
TZ and ZW conceived the study. TZ, ZW, JW, WF conducted the laboratory measurements. TZ carried
out the data analysis. TZ, KB, YY, XY, ZB, XM, YZ participated in the instrument managements. SG,
YC, CL, YZ, S-ML and MH supported this research. TZ wrote the paper with inputs from all co-authors.
**Competing interest**
The authors declare that they have no known competing financial interests or personal relationships that
could have appeared to influence the work reported in this paper.





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
