# Peer review of "A new smog chamber system for atmospheric multiphase"

_Atmospheric Measurement Techniques, 2023_

## Author Comment (AC1)

We thank reviewers for reviewing our manuscript and also for useful hints and suggestions. Below, comments from the reviewer are given in blue while our answers are given in black, with passages including new text given in red.
* * *
RC1: 'Comment on amt-2023-34', Anonymous Referee #2, 26 Apr 2023:

The manuscript by Zong et al., titled "A new smog chamber system for atmospheric multiphase chemistry simulation: design and characterization" characterized the 2 m³ indoor smog chamber (AIR) and conducted a series of characterization experiments. The characterization results, as well as the yield experiments of α-pinene ozonolysis combined with a box model simulation, supported that the AIR chamber could be used to simulate multiphase atmospheric chemical reactions. The manuscript was well written and organized, but the reviewer thinks that detail characterizations related to new designs for multiphase chemistry may be further strengthened.

Specific comments:

1)    Figure 2: the conditions of the test should be listed, such as the fan condition. Is it batch mode? Will irradiation affect the temperature? Can water molecules permeate the FEP film? If so, the RH should decrease with time due to the water permeation.

**Response:**

This chamber system is designed to operate in Batch Mode. The fans are only turned on during the injection period and kept off after finishing injection. The irradiation can change the temperature range of stable temperature control, but it does not affect the stability of temperature control. This is mentioned in Section 3.3: 'The illumination of lamps raises the lowest achievable temperature by 3 °C for every 10 lights on. However, the illumination of the reactor does not affect the stability of temperature and RH inside the reactor'.

FEP film is semi-permeable, and gaseous molecules can penetrate through it, but at a slow rate. For example, the permeation rate of nitrogen is about 5L/m²/24h/atm, while that of water molecules is only, 0.007L/m²/24h/atm. For batch mode experiments, which typically last around 3 hours, this permeation amount can be ignored. In fact, the stable RH observed in Figure 2 also suggests this point. These details were clarified in the revised manuscript.

**Modification in the main text:**

Line 374 – 375 (the description of Figure 2): added 'The chamber was operated in batch mode';

Line 353 – 355, added 'The RH fluctuation caused by the water permeation through the FEP filter can be ignored due to the slow permeation rate of water molecules (0.007L/m2/24h/atm)'.

Line 384 – 386, added the discussion about the fans 'In Table S6, when turn on the fans, the wall loss is usually much higher, so the fans will only be turned on during the injection period and kept off after injection'.

2)    What is the size distribution of the particles before and after condensing organics on the particles? Is it efficient to introduce monodisperse seeds (number concentration and size

**Response:**

In coating tests, the homogeneous nucleation of organic vapors after cooling is related to several factors such as the volatility of organic species, the number concentration/surface area concentration of the existing seed particles, and the cooling rate. As one part of the chamber system, the detailed operating conditions of the coating device need to be carefully considered and tested before each set of experiments to obtain monodisperse particles with coating morphology. For the results shown in Figure 3b, we have optimized the operating conditions and achieved monodisperse coating particles without nucleation of organic vapors.

The size distribution of aerosols was measured by SMPS, and two images are attached below for comparison, where the left graph shows the size distribution of 200nm monodisperse NaCl particles that have not passed through the coating device, and the right graph shows the size distribution of aerosols generated after passing through the coating device and forming organic coatings. This suggests that the monodisperse seeds are sufficient and no homogeneous nucleation of organic vapors.

Regarding the homogeneous nucleation of organic vapors in the coating device, we will add relevant statements in the revised manuscript for clarification.

[Figure]

**Modification in the main text:**

Line 414 – 417: added 'The surface area concentrations of the introduced seed ($> 800\ \mu m^2/cm^3$) are sufficient that no homogeneous nucleation of organic vapor occurs. Both the size distributions of the particles before and after condensing organics on the particles are monodisperse'.

3) For the pinene ozonolysis experiments, is there homogeneous nucleation with the high concentrations of pinene and ozone. I think the experimental conditions should be optimized for multiphase chemistry. For example, SOA is generated on the seed surfaces as much as possible.

**Response:**

During the preliminary experiments, a nucleation-growing particle population did exist due to the insufficient surface area concentration of seed particles. We did not use the data from these preliminary experiments. Later, we increased the loading of seed particles to avoid this situation. The corresponding time series of particle size distribution can be found in Figure S11 in SI, which supports this point.

**Modification in the main text:**

Line 467 – 468: added 'We used seeds with sufficient surface area concentration to prevent the gas phase products of VOC from homogeneous nucleation'.

4) Section 4: The vapor wall loss can have significant influence on SOA formation. How this will affect the SOA formation and SOA yield in this study, especially for different systems in the absence or presence of seed aerosols?

**Response:**

Quantifying wall losses of gaseous organic products is still a challenge in chamber experiments. Gaseous intermediates are difficult to be quantitatively measured, and the theoretical calculations of wall losses also have large uncertainties due to the lack of data on some parameters, such as the effective wall mass concentration and eddy diffusion coefficient inside the reactor. Therefore, we cannot determine the specific role of wall losses of gas-phase products in the formation and yield of SOA. The wall loss behavior of gases essentially depends on the concentration gradient between the gas phase and the wall. Compared with experiments without seed particles, when seed particles exist, gases condense on the particles while condensing on the walls, causing the gas-phase concentration to decay more rapidly, resulting in less wall loss of gases and higher SOA yields during the initial period of the experiment. However, the extent of this difference is still unclear because, under the condition without seed particles, particles generated through nucleation continue to grow and can provide a considerable amount of condensation sink after the reaction proceeds for a period of time. This process needs to be described and analyzed with models that carefully consider wall loss behavior and physicochemical properties of particles, but to our knowledge, it seems that such models are lacking and need to be established first. We are currently working on this task.

**Modification in the main text:**

Line 484 – 501: added the paragraph 'It is worth noting that, the organic vapor wall loss can have significant influence on SOA formation. However, quantifying wall losses of gaseous organic products is still a challenge in chamber experiments. Gaseous intermediates are difficult to be quantitatively measured, and the theoretical calculations of wall losses also have large uncertainties due to the lack of data on some parameters, such as the effective wall mass concentration and eddy diffusion coefficient inside the reactor. The wall loss behavior of gases essentially depends on the concentration gradient between the gas phase and the wall. To our knowledge, there is no conclusive evidence to support higher wall losses of gaseous intermediates under higher RH, which are even significant enough to cause a notable reduction in SOA yield. In addition, if higher RH can enhance the diffusion of gaseous intermediates towards the wall, then the diffusion of gaseous intermediates towards the particle phase should also increase. Compared with experiments without seed particles, when seed particles exist, gases condense on the particles while condensing on the walls, causing the gas-phase concentration to decay more rapidly, resulting in less wall loss of gases and higher SOA yields during the initial period of the experiment, as shown in the subplot of Fig. 5. However, the final difference in SOA yields is still unclear because, under the condition without seed particles, particles generated through nucleation continue to grow and can provide a considerable amount of

condensation sink after the reaction proceeds for a period of time. This process needs to be numerically described and analyzed that carefully consider wall loss behavior and physicochemical properties of particles in future studies'.

---

## Author Comment (AC2)

We thank reviewers for reviewing our manuscript and also for useful hints and suggestions. Below, comments from the reviewer are given in blue while our answers are given in black, with passages including new text given in red.

\*\*\*\*\*\*\*\*\*\*\*\*\*\*\*\*\*\*\*\*\*\*\*\*\*\*\*\*\*\*\*\*\*\*\*\*\*\*\*\*\*\*\*\*\*\*\*\*\*\*\*\*\*\*\*\*\*\*\*\*\*\*\*\*\*\*\*\*\*\*\*\*\*\*\*\*\*\*\*\*\*

RC2: 'Comment on amt-2023-34', Anonymous Referee #1, 03 May 2023

The authors show their results about a newly constructed smog chamber for multiphase chemistry studies. The facility and characterization results have been demonstrated. This work is well organized and presented. It is publishable after the following questions have been well addressed.

Specific comments:

1) The novelty of this work should be demonstrated based on a good review of previous studies.

**Response:**

We have checked the Introduction Section and found that the presentation and discussion of the application of chamber in atmospheric multiphase chemistry research is a little weak, so we have made some modifications to the relevant paragraph after investigating more previous studies.

Besides, we found that the last part of the current introduction only briefly summarizes the main work of this study without emphasizing the advantages of the AIR chamber system. We have now included this in our revised manuscript.

**Modification in the main text:**

Line 95 – 96: added 'In addition, small chambers may have the potential for controlling RH change and simulating co-condensation phenomena';

Line 98 – 108: rewrite as 'The influence of aerosol phase state on kinetics of gas-particle interactions has received increasing attention (Virtanen et al., 2010; Berkemeier et al., 2016; Wang et al., 2015a; Reid et al., 2018), and this requires the phase state of seed particles can be controlled in chamber simulations. A laboratory study using pre-deliquescence way to control particle phase state has been reported (Faust et al., 2017), providing a feasible way for phase state control. In regard of particle morphology, some chamber-based experimental studies in recent years have preliminarily shown that organic coatings have important effects on the kinetics of aerosol multiphase transformation (Zhou et al., 2019; Zhang et al., 2018; Zhang et al., 2019), which deserves more researches. As these studies showed evidence that the morphology and phase state of aerosol particles play important roles in the atmospheric multiphase chemistry processes, focused chamber studies on multiphase chemistry require additional steps to control the morphology and phase state of seed particles in chamber design';

Line 119 – 122 (last part of Introduction): added 'Our results indicate that the AIR chamber system has more precise temperature and RH control capabilities compared to other chambers. Phase state and morphology of seed particles can also be accurately manipulated in advance, which is rare in existing smog chamber systems'.

2) Line 105, the uncertainty of RH measurement is ±75%. What is the uncertainty for the RH

**Response:**

±0.75%, refers to the standard deviation of measured RH data sets, which deviate from the mean value while controlling the chamber RH at a set value, that within 0.75 of an RH unit (RH is described in %). Your statement 'the uncertainty for the RH sensor' should refer to the extent to which measured RH values deviate from the actual RH. We calibrated the RH sensor using a hygrometer, and the deviation is less than 1%, as explained in section 3.3 of the manuscript (Line 349).

3) Lines 223-225, is the coating system stable enough for generating organic coated particles? In my experience, the coating efficiency might decrease with time at high temperatures.

**Response:**

Yes, we did observe such phenomenon when testing the operation conditions of the coating system. The coating efficiency would drop after running continuously for approximately four hours. We noticed that if the generated inorganic core seeds were not thoroughly dried, condensed water would appear on the bottom of the three-necked flask after several hours, submerging the organics and resulting in a reduction of coating efficiency. Increasing the drying efficiency has alleviated the issue of reduced coating efficiency, allowing the stability of the coating system to last beyond four hours. In fact, for general chamber experiments, the step of injecting seeds will not last four hours, so the stability of this coating device is sufficient.

**Modification in the main text:**

Line 240 – 241: added 'The coating efficiency can keep stable within four hours, which is sufficient to meet the duration of injecting seeds for general experiments'.

4) How long does it take for introducing around 5000 cm$^{-3}$ OA-coated seeds into the chamber?

**Response:**

This depends on factors such as the size distribution of generated core seeds from the aerosol generator, the selected particle size, and the aerosol flow rate. For the generator we used, TSI 3076, the generated aerosol size distribution is not wide, and the peak particle size is between 50~100 nm. We tested that when the aerosol generation efficiency was set to its maximum, the concentration of monodisperse nanoparticles in the reaction chamber reached 5000 cm$^{-3}$ within a few seconds for 100 nm particles, about 5 minutes for 150 nm particles, and 40~50 minutes for 200 nm particles. It should be noted that the operating curve of the coating device is variable, depending on the operation goal such as injection concentration, and needs to be calibrated before formal experiments.

**Modification in the main text:**

Line 211 – 216: added 'For polydisperse seeds experiments, the seed generation system can inject seeds into the reactor to the desired amount within a time scale of seconds to several minutes. For monodisperse seeds experiments, if large-sized seeds that have a lower number fraction in the generated aerosol population are selected, the time scale will expand to 40-50 minutes. The appropriate time scale for seed injection can be adjusted by changing the solution concentration and aerosol flow rate'.

5)	Figure 4, why is the SOA yield higher than those reported values at high OA mass loading?

**Response:**

There are two reasons for this. One is that the experimental temperature we used is 293K, slightly lower than the temperatures used in the literatures. The second reason is that, as mentioned in Section 3.4, the Teflon film of our reaction chamber has been passivated by $O_3$, so the wall losses of intermediate products from the gas-phase reactions may be lower due to less reaction loss.

**Modification in the main text:**

Line 452 – 453: added 'and the lower experimental temperature'.

6)	When calculating SOA yields, the wall loss of LVOCs should be accounted for. The low SOA yield might be related to the high wall loss rate of LOVCs.

**Response:**

Quantifying wall losses of gaseous organic products is still a challenge in atmospheric chemistry research. Gaseous intermediates are difficult to be quantitatively measured, and the theoretical calculations of wall losses also have large uncertainties due to the lack of data on some parameters, such as the effective wall mass concentration and eddy diffusion coefficient inside the reactor. To our knowledge, there is no conclusive evidence to support higher wall losses of LVOCs under higher RH, which are even significant enough to cause a notable reduction in SOA yield. In addition, according to gas phase diffusion theory, if higher RH can enhance the diffusion of LVOCs towards the wall, then the diffusion of LVOCs towards the particle phase should also increase.

We still suggest attributing the lower SOA yield to higher liquid water content of the aerosol, and similar phenomena have also been observed in several groups of experiments in our another research after several months. We are currently attempting to obtain an explanation through theoretical calculations and have preliminarily found that this is related to the kinetic limitations of the particle phase dynamics, but which is different from the previously recognized viscosity and diffusion coefficient influences.

**Modification in the main text:**

Line 484 – 501: added the paragraph 'It is worth noting that, the organic vapor wall loss can have significant influence on SOA formation. However, quantifying wall losses of gaseous organic products is still a challenge in chamber experiments. Gaseous intermediates are difficult to be quantitatively measured, and the theoretical calculations of wall losses also have large uncertainties due to the lack of data on some parameters, such as the effective wall mass concentration and eddy diffusion coefficient inside the reactor. The wall loss behavior of gases essentially depends on the concentration gradient between the gas phase and the wall. To our knowledge, there is no conclusive evidence to support higher wall losses of gaseous intermediates under higher RH, which are even significant enough to cause a notable reduction in SOA yield. In addition, if higher RH can enhance the diffusion of gaseous intermediates towards the wall, then the diffusion of gaseous intermediates towards the particle phase should also increase. Compared with experiments without seed particles, when seed particles exist, gases condense on the particles while condensing on the walls, causing the gas-phase concentration to decay more rapidly, resulting in less wall loss of gases and higher

SOA yields during the initial period of the experiment, as shown in the subplot of Fig. 5. However, the final difference in SOA yields is still unclear because, under the condition without seed particles, particles generated through nucleation continue to grow and can provide a considerable amount of condensation sink after the reaction proceeds for a period of time. This process needs to be numerically described and analyzed that carefully consider wall loss behavior and physicochemical properties of particles in future studies'.

7) I am wondering why the authors do not evaluate the photochemical performance here.

**Response:**

The photochemical reaction is an important issue of atmospheric gas phase chemistry, which includes the study of the effects of VOCs, gas-phase oxidants, radiation, etc. However, the design of this chamber system mainly focuses on the physical and chemical properties of seed particles, so a relatively simple gas phase chemistry reaction system was chosen. For example, we adjust the phase state and liquid water content of seed particles by changing RH, but for a photochemical reaction system, the oxidant ·OH would be strongly affected by RH, leading to the roles of particle physical and chemical properties in SOA formation becoming ambiguous. Therefore, for the photochemical behavior of this chamber system, it is mainly characterized by spectral features and photolysis rate constants in Section 3.2, and no specific oxidizing system for photochemical reactions is conducted.

---

## Author Comment (AC3)

We thank reviewers for reviewing our manuscript and also for useful hints and suggestions. Below, comments from the reviewer are given in blue while our answers are given in black, with passages including new text given in red.

\*\*\*\*\*\*\*\*\*\*\*\*\*\*\*\*\*\*\*\*\*\*\*\*\*\*\*\*\*\*\*\*\*\*\*\*\*\*\*\*\*\*\*\*\*\*\*\*\*\*\*\*\*\*\*\*\*\*\*\*\*\*\*\*\*\*\*\*\*\*\*\*\*\*\*\*\*\*\*\*\*\*\*\*\*\*\*\*\*

RC3: 'Comment on amt-2023-34', Anonymous Referee #3, 11 May 2023

This study designed and characterized a new smog chamber that can be used to study atmospheric multiphase chemistry. The smog chamber system can achieve precise control of temperature and humidity to generate seed particles at different phases states. Some key parameters for the new chamber, including leak proofness, mixing performance, the wall losses for gas and particle phases etc, were systematically characterized. The new smog chamber system shows its ability to simulate secondary aerosol formation and atmospheric multiphase processes like other chambers. Overall, the manuscript is well organized. Some questions shall be clarified before publication:

**Major comments:**

1)    Line 138-140: Have the authors measured the temperature differences in different positions of the chamber? The sensor was set in the bottom now. How about the spatial distribution of the sensor.

**Response:**

We did not measure the spatial distribution of temperature. On the one hand, the internal space of the chamber is quite small, which is not suitable to arrange many temperature sensors. On the other hand, it is unnecessary. The internal design of our chamber enclosure is that, the circulating air for temperature control surrounding the reactor, and the temperature ultimately reaches equilibrium from the outside of Teflon film to the inside. The temperature control design of the chamber system reported in Wang et al (2014) is similar to ours, and their reactor is 30 $m^3$, with a spatial distribution difference in temperature within $\pm$ 0.5K. Our reaction chamber is only 2 $m^3$, and the spatial difference in temperature can be ignored.

**Modification in the main text:**

Line 344 – 348: added 'The internal design of our chamber enclosure ensures that the circulating air, which controls the temperature surrounding the reactor, reaches equilibrium (taking < 2 hours as shown in Fig. 2) from the outside of the Teflon film to the inside. This design guarantees that the temperature distribution is spatially homogeneous, even for a chamber system with a 30 $m^3$ reactor (Wang et al., 2014)'.

2)    Line 442-443: Are there any other studies about NaCl seed found the same conclusion? Is the dry NaCl seed not competitive with sorption of organic vapors onto the chamber wall? Or the condensing SOA compounds form a separate phase from the seed?

**Response:**

The seeds we used in the α-pinene ozonolysis experiments were ammonium sulfates, which was clarified in Section 4.2. This was selected to allow comparison with previous chamber studies since ammonium sulfate is the most commonly used seed while sodium chloride is relatively less used. In terms of SOA yield in the α-pinene ozonolysis, we found that in chamber studies, liquid water

tends to inhibit its formation (Cocker Iii et al., 2001; Kristensen et al., 2014), while in oxidation flow reactor studies, it tends to enhance the SOA yield (Faust et al., 2017; Zhao et al., 2021). Unfortunately, to our knowledge, these studies did not analyze the reasons behind this phenomenon in detail. In a subsequent study a few months later, we observed the same phenomenon and have preliminarily focused on the kinetic limitation of liquid water, but further theoretical calculations are still needed to explain it in detail.

Dry seeds still compete with the sorption of organic vapors onto the chamber wall, and in gas-particle partitioning, this is related to the vapor pressure of organic species in the condensed phase, and for particulate phase, curvature effect should be considered. Dry seeds have relatively higher vapor pressure of organic materials due to their lower Raoult effect, so they are not conducive to SOA formation in the initial stage of the reaction, as shown in the subplot of Figure 5. However, as SOA continues to form, the difference in SOA yield reverses.

The homogenous nucleation of organic vapor was avoided by increasing the seed number concentration/surface area concentrations. The particle size distribution in Figure S11 demonstrates this.

**Modification in the main text:**

Line 484 – 501: added the paragraph 'It is worth noting that, the organic vapor wall loss can have significant influence on SOA formation. However, quantifying wall losses of gaseous organic products is still a challenge in chamber experiments. Gaseous intermediates are difficult to be quantitatively measured, and the theoretical calculations of wall losses also have large uncertainties due to the lack of data on some parameters, such as the effective wall mass concentration and eddy diffusion coefficient inside the reactor. The wall loss behavior of gases essentially depends on the concentration gradient between the gas phase and the wall. To our knowledge, there is no conclusive evidence to support higher wall losses of gaseous intermediates under higher RH, which are even significant enough to cause a notable reduction in SOA yield. In addition, if higher RH can enhance the diffusion of gaseous intermediates towards the wall, then the diffusion of gaseous intermediates towards the particle phase should also increase. Compared with experiments without seed particles, when seed particles exist, gases condense on the particles while condensing on the walls, causing the gas-phase concentration to decay more rapidly, resulting in less wall loss of gases and higher SOA yields during the initial period of the experiment, as shown in the subplot of Fig. 5. However, the final difference in SOA yields is still unclear because, under the condition without seed particles, particles generated through nucleation continue to grow and can provide a considerable amount of condensation sink after the reaction proceeds for a period of time. This process needs to be numerically described and analyzed that carefully consider wall loss behavior and physicochemical properties of particles in future studies'.

Line 466 – 467: added 'We used seeds with sufficient surface area concentration to prevent the gas phase products of VOC from homogeneous nucleation'.

3)    Line 372-373: How to determine the segmentation of the shift and whether the results of the shift accurately represent the actual wall loss of the substance in different experiments?

**Response:**

The determination of the shift points can be roughly determined by human judgment. We had the curve of the wall loss rate constant β as a function of particle size Dp for each measured particle diameter, which allows us to visually identify two shift points. Then, we observed the corresponding particle size bin in the raw data where β changes. For instance, as Dp increases, β first increases and reaches a maximum value at 45.3 nm. Starting from the next particle size bin at 52.3 nm, β decreases with increasing Dp. Therefore, the first shift point is determined to be between 45.3 nm and 52.3 nm, and the value can be taken as 50 nm. So was the identification of another shift point

As for the second question, there may be some differences in β for identical Dp in each experiment, but the two shift points remain the same. Therefore, in the formal experiments using monodisperse seeds, a 20-30 minute period is reserved before each reaction to determine β for this monodisperse particle size in that particular experiment. Then, the β for other particle sizes can be determined by an up-and-down shift of the $\log_{10}(k)$-$\log_{10}(Dp)$ function curve (as mentioned in Section 3.4). For experiments with polydisperse seeds, the total volume wall loss rate constant (mentioned in the last paragraph of Section 3.4) is typically used, and this parameter also needs to be determined 20-30 minutes before each experiment.

**Modification in the main text:**
Line 394 – 396: changed 'In this study, two inflection points are identified at 50 nm and 150 nm (Fig. S8)' as 'In this study, two inflection points are selected at 50 nm and 150 nm according to the identified inflection particle size bin of 45.3 – 53.2 nm and 143.3 – 165.5 nm, respectively (Fig. S8)'.

**Technical comments:**

4)   Line 239-240: Please note the use of subscripts ($SO_2$ and $O_3$).

**Response:**
Thank you, we have corrected it now.

**Modification in the main text:**
Line 255 – 256: changed the subscripts ($SO_2$ and $O_3$).

5)   The description of figures should be more accurate:

Line 259-261: "more than 25 hours" could not be seen in Fig. S2a.

Line284-285: Most of the mixing time to uniformity for gas showed in the figure is longer than 1minute with the fans on.

**Response:**
Thank you for your correction. We have changed it to 'within 24 hours' for more precise description.

We apologize for any potential confusion caused by the left panel of Fig. S5, but most mixing times are indeed within one minute. We have added a sentence in the main text to clarify: 'the duration between the two plateaus in Fig. S5'.

**Modification in the main text:**

Line 276: changed 'more than 58 hours' as 'within 24 hours'.

Line 300 – 301: added '(the duration between the two plateaus in Fig. S5)'.

6) Line 321-323: The number of lights of scheme 'left and right' is 20 but not 40, and is 10 lights for 'only left' and 'only right'.

**Response:**

Thank you for your correction, we have changed.

**Modification in the main text:**

Line 338 – 339: changed the correct numbers of lights: 'the scheme 'left and right' (20 lights) was the sum of that of 'only left' (10 lights) and 'only right' (10 lights)'.

7) Line 378-380: Please note the use of operational symbol (* and Í).

Line 198: The format of unit shall be checked through, e.g., should be no 'space' before %. The same unit with different format such as cm-3 and μg/m3 were used.

**Response:**

Thank you for your correction, we have changed the symbol '*' as '×' (Line 386) and '·' (Line 361).

**Modification in the main text:**

Line 404: changed the symbol '*' as '×';

Line 379: changed the symbol '*' as '·';

Line 446: changed the symbol '*' as '·'.

8) Please unify the format of the graphs and optimize the graphs. For example, the font size is different in Fig. 3a and 3b. There should be a 'space' between 20 and °C for Fig. 3b. And we could even see some grey lines outside the graphs, it seems that the graphs were simply pasted from other software and combined together. Please redraw these figures and unify the format.

**Response:**

Thank you for your suggestion, we have unified the format of the graphs and optimized the graphs.

**Modification in the main text:**

We have optimized Figure 3, and unified the format of Figure 2 to Figure 5.

9) Please doublecheck the citation of references. For example,

Line 85, the citation of "Ravishankara, 97" should be 1997 but not 97.

Line 762-764, the year was missing for this citation.

**Response:**

Thank you for your reminding, we have double checked and corrected the improperly formatted citations.

**Modification in the main text:**

Line 85: corrected as '1997';

Line 540, 554 – 555, 557, 559, 573 - 574, 578, 587, 655, 692, 694, 704, 752, 784, 812: completed the year of these citations.

**Reference:**

Cocker Iii, D. R., Clegg, S. L., Flagan, R. C., and Seinfeld, J. H.: The effect of water on gas–particle partitioning of secondary organic aerosol. Part I: α-pinene/ozone system, Atmospheric Environment, 35, 6049-6072, https://doi.org/10.1016/S1352-2310(01)00404-6, 2001.

Faust, J. A., Wong, J. P. S., Lee, A. K. Y., and Abbatt, J. P. D.: Role of Aerosol Liquid Water in Secondary Organic Aerosol Formation from Volatile Organic Compounds, Environmental Science & Technology, 51, 1405-1413, 10.1021/acs.est.6b04700, 2017.

Kristensen, K., Cui, T., Zhang, H., Gold, A., Glasius, M., and Surratt, J. D.: Dimers in alpha-pinene secondary organic aerosol: effect of hydroxyl radical, ozone, relative humidity and aerosol acidity, Atmospheric Chemistry And Physics, 14, 4201-4218, 2014.

Wang, X., Liu, T., Bernard, F., Ding, X., Wen, S., Zhang, Y., Zhang, Z., He, Q., Lü, S., Chen, J., Saunders, S., and Yu, J.: Design and characterization of a smog chamber for studying gas-phase chemical mechanisms and aerosol formation, Atmospheric Measurement Techniques, 7, 301-313, 10.5194/amt-7-301-2014, 2014.

Zhao, R. R., Zhang, Q. X., Xu, X. Z., Zhao, W. X., Yu, H., Wang, W. J., Zhang, Y. M., and Zhang, W. J.: Effect of experimental conditions on secondary organic aerosol formation in an oxidation flow reactor, Atmos Pollut Res, 12, 392-400, 10.1016/j.apr.2021.01.011, 2021.